# Pressure Injury Prediction in Intensive Care Units Using Artificial Intelligence: A Scoping Review

**DOI:** 10.3390/nursrep15040126

**Published:** 2025-04-09

**Authors:** José Alves, Rita Azevedo, Ana Marques, Rúben Encarnação, Paulo Alves

**Affiliations:** 1Center for Interdisciplinary Research in Health, Faculty of Health Sciences and Nursing, Universidade Católica Portuguesa, 4169-005 Porto, Portugal; s-arrazevedo@ucp.pt (R.A.); s-anjomarques@ucp.pt (A.M.); rcencarnacao@ucp.pt (R.E.); pjalves@ucp.pt (P.A.); 2Intensive Care Unit, Braga Local Healthcare Unit, 4710-243 Braga, Portugal; 3Intensive Care Unit, Gaia and Espinho Local Healthcare Unit, 4434-502 Vila Nova de Gaia, Portugal; 4Cardiology Intensive Care Unit, São João Local Healthcare Unit, 4200-319 Porto, Portugal

**Keywords:** artificial intelligence, pressure injury, intensive care units, critical care, critical care nursing

## Abstract

**Background/Objetives:** Pressure injuries pose a significant challenge in healthcare, adversely impacting individuals’ quality of life and healthcare systems, particularly in intensive care units. The effective identification of at-risk individuals is crucial, but traditional scales have limitations, prompting the development of new tools. Artificial intelligence offers a promising approach to identifying and preventing pressure injuries in critical care settings. This review aimed to assess the extent of the literature regarding the use of artificial intelligence technologies in the prediction of pressure injuries in critically ill patients in intensive care units to identify gaps in current knowledge and direct future research. **Methods:** The review followed the Joanna Briggs Institute’s methodology for scoping reviews, and the study protocol was prospectively registered on the Open Science Framework platform. **Results:** This review included 14 studies, primarily highlighting the use of machine learning models trained on electronic health records data for predicting pressure injuries. Between 6 and 86 variables were used to train these models. Only two studies reported the clinical deployment of these models, reporting results such as reduced nursing workload, decreased prevalence of hospital-acquired pressure injuries, and decreased intensive care unit length of stay. **Conclusions:** Artificial intelligence technologies present themselves as a dynamic and innovative approach, with the ability to identify risk factors and predict pressure injuries effectively and promptly. This review synthesizes information about the use of these technologies and guides future directions and motivations.

## 1. Introduction

Pressure injuries (PIs) are a cross-cutting problem in various healthcare contexts, with an important impact on patients and healthcare systems [1,2,3,4]. The development of PIs can lead to reduced quality of life, worsening pain, risk of infection, and increased length of hospital and intensive care stays, and is also associated with increases in hospital readmission and mortality rates. It thus represents an important increase in direct and indirect costs related to healthcare [3,4,5].

A PI can be defined as a localized injury to the skin and/or underlying tissues as a result of pressure or a combination of pressure and torsion forces, usually located in areas of bony prominence [6]. The injury arises due to the forces exerted by the individual’s body weight or because of external forces such as those applied by a medical device or other object, or a combination of these. Tissue damage occurs as a result of prolonged and sustained exposure to compression deformations (perpendicular to the surface of tissues), tension, torsion (parallel to the surface of tissues), or a combination of both [6,7].

These lesions are often considered preventable, but they are still a problem with a relevant prevalence at the hospital level. Systematic reviews on the global incidence and prevalence of PIs in hospitalized adult patients report a prevalence ranging from 12.8% to 14.8%, with a pooled hospital-acquired PI rate of 6.3% to 8.4% [8,9].

Intensive care units (ICUs) are characterized by highly differentiated hospital units that provide continuous support and monitoring to people in critical and acute conditions [10]. Factors such as multiorgan failure, hemodynamic instability, inadequate perfusion and oxygenation, multiple comorbidities, reduced mobility, specific medication and insufficient nutritional support are related, in this context of care, to a significant increase in the risk of developing PIs and consequent increases in the prevalence of the phenomenon [11,12]. Chaboyer et al. [11] estimated a cumulative prevalence of PIs in ICUs of 16.9–23.8% and a mean incidence of 10.0–25.9%. These results have recently been reinforced by a large-scale prospective observational study conducted in 1117 ICUs in 90 countries, in which data were collected from 13254 individuals. This revealed an overall prevalence of PIs in ICUs of 26.6%, and the prevalence of PIs acquired in intensive care was 16.2% [13].

Considering PIs as an adverse event that are often preventable and used as an indicator of the quality of nursing care, it is essential to identify individuals at risk correctly and effectively. Risk identification is the starting point for implementing appropriate preventive measures and effectively managing the available resources [14,15,16].

Coleman et al. [14] proposed a conceptual framework that categorizes PI risk factors along causal pathways, encompassing intrinsic factors (e.g., perfusion, nutrition, and mobility) and extrinsic factors (e.g., skin moisture, shear, and pressure). This model highlights the multifactorial and dynamic nature of PI risk, underscoring the need for predictive tools capable of integrating a broad range of clinical variables.

To this end, several risk assessment instruments have been developed, such as the Norton [17], Braden [18] and Waterlow [19] scales. These scales consider general dimensions that lead to the development of PIs but do not consider some relevant risk factors such as hematological values, oxygenation, perfusion, and certain comorbidities such as diabetes and vascular pathology. Risk factors for specific populations, such as patients hospitalized in ICUs, are not yet considered [12]. This fact means that, when applied in this context, they have low predictive power, with high sensitivity, but with low specificity, and thus low discriminatory power, as reported in several studies [20,21,22,23,24,25].

In response to this problem, specific instruments have been developed for the intensive care context, such as the Cubbin and Jackson [26], CALCULATE [25,27] and EVARUCI [28] scales, which have greater predictive power in this patient population. Even so, they are static instruments, applied punctually by an operator, without producing a dynamic, real-time response to the patient’s health status.

In addition to their structural limitations, the performance of traditional risk assessment tools is influenced by the clinician’s level of experience and subjective judgment. Variations in clinical interpretation can lead to significant inter-observer variability, compromising the consistency and reliability of risk classification. In intensive care settings, where the timely and accurate identification of PI risk is crucial, such variability may delay preventive interventions or result in inappropriate resource allocation. Even with tools specifically developed for critical care populations, dependence on manual and experience-based assessment remains a limitation to standardized, evidence-based decision-making [29].

Derived from the increasing availability of Electronic Health Records (EHRs) data, important advances have been made in the development of new technological instruments that allow the development of new solutions to this problem, namely, artificial intelligence (AI). AI is a system or machine simulation of human intelligence [30]. This concept includes machine learning (ML) models, deep learning, neural networks, and natural language processing.

Data mining techniques are key for predicting phenomena, enabling the extraction and categorization of large datasets from EHRs. ML models, a subset of AI, allow systems to learn from data and build predictive models [31]. These approaches reveal complex patterns in EHR data, identifying relationships between variables to predict outcomes like PIs [32,33,34]. Unlike traditional scales, ML models analyze a broader range of risk factors, calculate the importance of each variable, and adapt autonomously over time as new cases are integrated into information systems [35].

The performance of ML classifiers is validated using metrics like receiver operating characteristic (ROC) curves and the area under the ROC curve (AUROC) [36,37]. A confusion matrix summarizes correct and incorrect classifications, while other metrics, such as the area under the precision–recall curve (AUPR), F1 score, accuracy, sensitivity, specificity, precision, and negative predictive value, offer deeper insights. Sensitivity measures the likelihood of correctly identifying a positive case, specificity reflects the ability to exclude negatives, and precision indicates the probability of disease presence when the test is positive. The AUROC quantifies a model’s discriminatory power, with values above 0.8 indicating robust and clinically reliable models, while those below 0.75 are considered inadequate for decision-making. This highlights AUROC’s importance as both a statistical measure and a benchmark for clinical applicability [36,37,38,39].

Current evidence reinforces the important role of nursing teams in addressing PIs. Thus, the development of more effective tools for the detection and stratification of individuals at risk has enormous potential, as well as profound implications, in the implementation of preventive measures and, more globally, in nursing practice.

A preliminary search was conducted in MEDLINE, SCOPUS, CINAHL (Cumulative Index to Nursing and Allied Health Literature), PubMed, Cochrane Database for Systematic Reviews, Joanna Briggs Institute (JBI) Evidence Synthesis, Web of Science Core Collection, PROSPERO and Open Science Framework databases, and no published or ongoing reviews were found on the use of AI to predict the development of PIs in critically ill patients admitted to ICUs. Two systematic reviews were identified that investigate the use of ML models in risk assessment in hospitals, but not in the specific context of intensive care [40,41], and another systematic review addresses AI technologies in risk management in ICUs but is not directed at the phenomenon of PIs [34].

Despite the growing interest in AI in healthcare, the current literature remains fragmented regarding its specific application to PI prediction in intensive care settings. Traditional risk assessment tools lack the ability to process complex, multidimensional data, and often fail to capture ICU-specific risk factors in real time. Although some reviews have examined AI in hospital-based risk prediction more broadly, they do not address the particular vulnerabilities of critically ill patients nor the technological approaches tailored to the ICU context. A focused synthesis of the available evidence is therefore essential to understand the current landscape, identify gaps, and inform the future development and implementation of AI models in this high-risk population.

Therefore, the purpose of this scoping review is to systematically map the existing evidence on the application of AI technologies in this context. Specifically, it aims to identify the types of AI models used, the variables involved in predictive modeling, the outcomes reported, and the implications for clinical and nursing practice.

To guide this review and ensure a comprehensive mapping of the available literature, the following research questions were formulated:What artificial intelligence tools are used in predicting the risk of pressure injuries in critically ill patients admitted to intensive care units?What are the results of using artificial intelligence tools in predicting pressure injuries in critically ill patients admitted to intensive care units?What variables are identified by artificial intelligence tools in predicting pressure injuries in critically ill patients admitted to intensive care units?What are the implications for nursing practice of using artificial intelligence tools in predicting pressure injuries in critically ill patients admitted to intensive care units?

## 2. Materials and Methods

This scoping review was conducted following The Joanna Briggs Institute guidelines for scoping reviews [42,43] and is reported following the recommendations of the Preferred Reporting Items for Systematic Reviews and Meta-Analyses Extension for Scoping Reviews (PRISMA-ScR) [44]. This study protocol was prospectively registered on the Open Science Framework platform (https://doi.org/10.17605/OSF.IO/5M3KH).

Accordingly, and in line with the stated objectives and research questions, this review aimed to identify and map the available scientific evidence on the use of AI in predicting PIs in critically ill individuals admitted to ICUs.

### 2.1. Eligibility Criteria

Participants: This review considered studies that include adult critically ill patients. No restrictions were applied regarding gender, ethnicity, or other personal characteristics. A critically ill person is defined as someone experiencing a critical illness, with a potentially reversible health condition characterized by vital organ dysfunction and a high risk of imminent death if appropriate care is not provided [45].Concept: Studies addressing AI for predicting PIs were considered. AI is understood as the simulation of human intelligence by a system or machine [30]. This concept includes, but is not limited to, ML, deep learning, neural networks, and natural language processing. Studies that address other types of instruments or tools will be excluded. A PI is an injury or ulceration caused by prolonged pressure on the skin and tissues when one stays in one position for a long period of time, such as lying in bed. Additionally, pressure injuries caused by medical devices, known as ‘medical device-related pressure injuries’, which typically develop in different locations than traditional PIs, will also be considered [6].Context: Regarding context, studies conducted in adult, specific, or multipurpose ICUs within public or private hospitals were included, without geographic or cultural limitations. Pediatric and neonatal ICUs were excluded. An ICU is an organized system for providing care to critically ill patients, offering intensive and specialized medical and nursing care, enhanced monitoring capabilities, and multiple modalities of physiological organ support to sustain life during acute organ system failure [10].

### 2.2. Types of Sources

This scoping review considered quantitative, qualitative, and mixed-methods studies. Literature reviews, dissertations and theses, text and opinion papers, books, and book chapters were also included.

### 2.3. Search Strategy

An initial exploratory search was conducted in CINAHL, MEDLINE and SCOPUS databases to identify synonymous search terms used in article indexing, titles, abstracts, and keywords.

The terms used for this preliminary search were “pressure ulcer”, “pressure injury”, “machine learning”, “artificial intelligence”, “deep learning”, “neural networks”, “critical care”, “intensive care units”, “intensive care”, “critical illness”, and “critically ill patients”. The free terms and indexing terms (Medical Subject Headings/CINAHL Subject Headings) identified were then used to develop the full search strategy for each database.

To identify relevant papers and documents for this review, both published and unpublished works were searched through electronic databases and grey literature. The electronic databases included CINAHL (by EBSCO), MEDLINE (by EBSCO), SCOPUS, PubMed, Cochrane Library (Cochrane Database for Systematic Reviews and Cochrane Central Register of Controlled Trials), Web of Science Core Collection, and Association for Computing Machinery Digital Library. For grey literature, searches were conducted in the Bielefeld Academic Search Engine (BASE) and Repositórios Científicos de Acesso Aberto de Portugal (RCAAP).

No language restrictions were applied, and no date limit has been set for publication, since the objective is to comprehensively assess the literature published on this topic. The electronic search was conducted by two independent reviewers (J.A and A.M) up to and including 31 March 2024, using the terms “pressure injury”, “artificial intelligence” and “intensive care units”, along with their related terms. The search strategy, including all identified keywords and index terms, was customized for each literature source (see Appendix A).

### 2.4. Source of Evidence Selection

The search strategy led to the identification of 137 titles. Their citations and abstracts were uploaded into Rayyan Web software [46] for bibliographic management and were organized according to the database from which they were retrieved. After the removal of duplicates, 72 papers were screened by title and abstract by two independent reviewers (J.A. and A.M.) with 93.1% agreement (k = 0.82), of which 16 were selected for full-text review. Of these, 10 met the inclusion criteria. The reference lists of these papers were then analyzed to identify additional papers of interest based on the relevance of their titles. Then, 7 papers were selected and assessed against the eligibility criteria, of which 4 met the inclusion criteria. Searches were also conducted on relevant organizations such as the European Pressure Ulcer Advisory Panel (EPUAP) and the National Institute for Health and Care Excellence (NICE) for relevant publications. Finally, 14 papers were included in this scoping review. The full texts retrieved were uploaded to Zotero software (Zotero, version 7.0.13) [47]. The process of paper identification, selection, eligibility, and inclusion is illustrated in the flow diagram below (Figure 1), as per Page et al. [48]. This process was conducted by two reviewers (J.A. and A.M.). Disagreements were resolved through consensus and, when necessary, consultation with a third reviewer (R.A.). During the selection process, a total of six reports were excluded based on the predefined criteria. One report was excluded because it involved participants outside the scope of the review, which focused on adult populations (n = 1). Two reports were excluded for not being relevant to the context of ICUs (n = 2), and three were excluded due to conceptual misalignment, as they did not address the use of AI for predicting PIs (n = 3). These exclusions ensured that the analysis included only studies strictly aligned with the review’s objectives.

### 2.5. Data Extraction

Data were extracted by two independent reviewers (J.A. and R.A.) using a data extraction tool developed in advance for this purpose. The extracted data include specific details about the studies, such as the title, authors, year of publication, country, source of information, objectives, study design, participants, setting, research tools, and the participants. Relevant results will also be extracted to answer the review questions, namely, the types of AI instruments used, their performance, the variables used, and the implications of their application for nursing practice.

The data extraction tool is the result of an adaptation of the data extraction tool proposed by the Joanna Briggs Institute [49]. Disagreements between the reviewers were resolved by consensus, or by consulting a third independent reviewer (A.M.). This scoping review did not seek to address highly specific research questions or evaluate the quality of the evidence generated; therefore, a critical appraisal of the methodological quality of the included papers was not conducted [50].

### 2.6. Ethical Considerations

Considering the nature of the study as a scoping review, which does not involve any participants, there are no ethical implications to be addressed.

## 3. Results

### 3.1. Characteristics of Included Papers

The papers included were published between 2013 and 2024, and the studies were carried out in the United States of America (n = 9), South Korea (n = 3), Spain (n = 1), and Czechia (n = 1). Over half (70%; n = 10) were published in the last 5 years. They primarily reported quantitative research. Most studies reported retrospective cohort studies (n = 11); one reported a two-phase retrospective and prospective cohort study, one reported a prospective cohort study, and one an experimental before-after design. All reviewed studies reported the development of different ML models based on EHR data for predicting PIs, namely, ensemble models, deep learning, neural networks, regression, Bayesian, decision trees, and instance-based. These terms are standardized as “prediction models” throughout this scoping review. The sample sizes varied between 27 and 74051 in studies designed as single or multicenter.

### 3.2. Prediction Model Design

The predictive models identified in this review were primarily developed using retrospective data extracted from EHRs. These structured datasets commonly included variables such as demographic characteristics, comorbidities, vital signs, laboratory values, medication administration, and nursing documentation. This indicates that data entry into the models was generally automated via EHR queries, rather than performed manually. While some studies employed deep learning methods capable of processing sequential ICU data [51], others used probabilistic approaches—such as Bayesian networks—to capture relationships between clinical variables [32,33].

The primary outcome across studies was the occurrence of hospital-acquired PIs, without limiting analysis to specific subtypes; some studies included stage I and above, while others focused on stage II or higher. This outcome variable was consistently obtained from manually documented clinical assessments in the EHR. Notably, no study reported the use of imaging technologies for visual wound scanning or the automated staging of PIs.

A total of 61 prediction models were developed for hospital-acquired pressure injury (HAPI) prediction, and the number of models in each of the 14 studies varied from 1 to 9 (Table 1). Eight studies used more than one model, and six studies used just one. The most common models used include logistic regression (n = 9), random forest (n = 8), and support vector machine (n = 5). The most commonly best-performing prediction models were logistic regression (n = 4), Bayesian networks (n = 2), and random forest (n = 2).

### 3.3. Variables

The studies included have identified key variables based on conceptual frameworks, such as Coleman et al.’s model [14], which classifies predictors along causal pathways. These studies selected variables, including immobility, skin status, and poor perfusion, as direct causal factors, supported by reviews of relevant literature, to align with PI etiology and enhance prediction approaches (full list of input variables for each study in Appendix A).

The studies analyzed a wide range of variables, which can be grouped into seven main categories:Demographics—age, gender, ethnicity, height, weight, and body mass index (BMI) [32,33,51,52,53,54,55,58,59,60]. These variables were widely used, serving as foundational data for risk analysis;Clinical Measures—Covers parameters such as blood pressure (systolic, diastolic, and mean), heart rate, oxygen saturation (SpO2), temperature, Glasgow Coma Scale, APACHE, and MEWS scores [21,32,51,53,54,58]. These measures play a key role in capturing the patient’s disease severity for hemodynamic and neurological conditions;Laboratory Results—Includes variables such as albumin, hemoglobin, glucose, creatinine, lactate, bilirubin, arterial PaO2, PaCO2, and pH [32,51,53,54,56,58]. These variables were often highlighted as significant predictors due to their ability to reflect the patient’s nutritional, metabolic and inflammatory status;Interventions—Variables related to clinical interventions, such as the use of ventilation (invasive or non-invasive), Continuous Renal Replacement Therapy (CRRT), Extracorporeal membrane oxygenation (ECMO), and parenteral nutrition [32,51,55,58,60]. These variables reflect the impact of therapeutic interventions on risk of injury development;Medication—Includes sedatives, vasopressors, analgesics, steroids, and diuretics [32,51,53,55,56,58];Nursing Assessments—Includes variables from the Braden scale (total score and subscales such as sensory perception, activity, mobility, nutrition, skin moisture, and friction/shear) [32,33,51,52,55,57,58,59,60], repositioning practices, and skin assessment (e.g., fragile skin or skin tears) [32,58,59];Comorbidities—Encompasses conditions such as diabetes, hypertension, heart failure, COPD, spinal cord injury, stroke, and cancer [21,33,51,52,54,60].

### 3.4. Model Performance

Table 2 summarizes the performance results of the best model for each study included in the review. The indicators used to measure the performances of the prediction models include the area under the receiver operating characteristic curve (AUROC), accuracy, sensitivity (SEN), specificity (SPE), positive predictive value (PPV) and negative predictive value (NPV). Other performance metrics used but not reported in more than 25% of the studies were not considered for this summary table, namely, the F1 Score, area under the precision–recall curve, true positive rate, true negative rate, and the Youden index (full table can be consulted in Appendix A).

In the 14 studies, 12 studies reported AUROC, ranging from 0.74 to 0.99; 4 studies reported accuracy, ranging from 0.87 to 0.96; 9 reported SEN, which ranged from 0.16 to 0.92; 7 reported SPE (0.69–0.99); 9 reported PPV (0.09–0.95); and 7 reported NPV (0.93–0.99).

### 3.5. Results of Implementation

Most studies included are modeling studies, which do not report any intervention effects. In their study, Ladíos-Martin et al. [21] report that the prediction model has been deployed into clinical practice and integrated with the EHR. According to the authors, the model enables nurses to accurately and objectively identify the risk of PI from admission to discharge. Recognizing changes in the patient’s condition over time helps caregivers concentrate on preventive care for those who need it most, without requiring nurses to collect new information. The data demonstrate that, when employed as a standalone assessment tool, the logistic regression model exhibits superior discriminant capability compared to the Norton scale (AUROC = 0.77 vs. 0.88). No other results were measured or reported in this study.

Cho et al. [32] employed a before-and-after experimental design. The prediction model was incorporated into a clinical decision support tool accessible to nurses in an ICU setting. The study’s results were measured regarding the prevalence of hospital-acquired PI and the length of stay in the ICU. The prevalence of hospital-acquired PI decreased from the baseline period (21%) to the intervention period (4.0%). The adjusted odds ratio for the intervention group relative to the baseline group was 0.1 (*p* < 0.0001), indicating a significant 10-fold decrease in HAPI prevalence. Additionally, the ICU length of stay decreased from 7.6 days in the baseline period to 5.2 days in the intervention period. The adjusted odds ratio for the intervention group versus the baseline group was 0.67 (*p* < 0.0001), indicating a significant 33% decrease in ICU length of stay.

## 4. Discussion

This review compellingly demonstrates the remarkable potential of AI to revolutionize the prediction of HAPIs, especially within ICUs. Drawing from an analysis of 14 pivotal studies, the findings powerfully highlight that ML-based predictive tools not only surpass traditional assessment methods like the Braden and Norton Scales, but also effectively address the limitations of these static approaches, which fail to account for the dynamic and complex nature of ICU patients’ conditions. Embracing AI in this context promises to enhance patient outcomes significantly and safeguard the well-being of those most vulnerable in healthcare settings.

Dynamic adaptability enables AI models to utilize real-time clinical data, such as vital signs, laboratory results, and patient-specific interventions, facilitating more accurate and timely predictions. Moreover, several studies have reported exceptional predictive performance, with AUROC values exceeding 0.90, highlighting their potential to enhance clinical decision-making and resource allocation. These advancements position AI as a crucial tool in proactive care strategies, particularly for high-risk populations in ICUs, where early detection and intervention are vital for improving patient outcomes and reducing the occurrence of HAPIs.

### 4.1. Performance Analisys of AI Models

Across the reviewed studies, several AI models demonstrated outstanding performance, with AUROC values ranging from 0.68 to 0.99, showcasing a broad spectrum of predictive accuracy across various models and clinical contexts. Notably, 10 out of the 14 studies reported adequate to exceptional discriminatory power, with AUROC values exceeding 0.8, highlighting the strong potential of these models in accurately identifying patients at risk. For instance, Šín et al. [60] reported an AUROC of 0.994 using a random forest model, making it one of the most accurate approaches in the review. Similarly, Kim et al. [51] achieved an AUROC of 0.945 using a Gated Recurrent Unit with decay (GRU-D++), a deep learning model adept at handling sequential data and time-dependent variables. These models stand out due to their ability to process complex, multidimensional datasets and adapt to the dynamic nature of ICU patient conditions.

The AUROC values in these studies are significantly higher than those typically reported for traditional methods like the Braden scale, which often falls below 0.8 in ICU settings [22]. However, some models have demonstrated improved performance when incorporating the Braden scale as an input variable. For example, Šín et al. [51] integrated the Braden scale with other clinical variables, reporting an AUROC of 0.83, suggesting that the Braden scale, while limited as a standalone tool, can provide meaningful contributions when used as part of a broader dataset. These findings underscore the potential of merging traditional risk assessment tools with AI methodologies to enhance the interpretative value of established practice scales.

Conversely, other studies found that including the Braden scale did not substantially enhance model performance. For instance, Cramer et al. [54] concluded that including the Braden scale in the model did not improve performance, as its subscales often show limited variability among ICU patients, making EHR-based models more effective and less dependent on time-consuming, subjective manual scoring. Also, in the study of Kim et al. [51] an analysis of the SHAP values revealed that the Braden scores were among the top ten most essential variables for predicting PI occurrence. However, predictions using only 42 variables (excluding the Braden score) resulted in a minimal 1.2–1.4% decrease in AUROC compared to all 48 variables, indicating that the 42 variables already encapsulate significant information about PI risk. This approach could reduce the nursing workload associated with calculating and recording the Braden score, making the model a promising tool for future application.

Other conventional risk assessment tools, such as the Norton scale, CALCULATE, and EVAR-UCI, while widely used in clinical practice, also generally demonstrate lower predictive accuracy when compared to advanced AI models. For instance, the Norton scale has shown inferior performance in ICU contexts, where patient conditions are more dynamic and complex when compared to ML models [21]. Similarly, CALCULATE and EVAR-UCI, though developed to address some ICU-specific challenges, rely on static, which may fail to capture the rapid changes in patient conditions. These conventional tools often prioritize ease of use and clinical applicability, but their performance metrics typically do not reach the thresholds achieved by modern AI models.

Despite its simplicity, logistic regression proved to be a robust and reliable model in several studies. Kaewprag et al. [52] and Ladíos-Martin et al. [21] reported logistic regression as their best-performing model, achieving AUROC values of 0.83 and 0.88, respectively. These results demonstrate the continued relevance of logistic regression, particularly in settings where interpretability and ease of implementation are prioritized. While logistic regression may not always match the precision of advanced models, its consistent performance across varied datasets and its transparency in clinical decision-making remain significant advantages.

However, not all models achieved optimal performance. Hyun et al. [55] used logistic regression and reported an AUROC of 0.74 with sensitivity and specificity of 0.65 and 0.69, respectively, indicating moderate predictive accuracy. The model struggled to balance these metrics, particularly in detecting true positive cases, highlighting the challenge of sensitivity in some AI applications. Similarly, Naïve Bayes models, as applied by Choi et al. [56], achieved an AUROC of 0.82, which, while acceptable, underscores their limitations when compared to more complex approaches like ensemble methods or neural networks.

The variation in model performance reflects the influence of several factors, including the choice of input variables, data quality, and sample size. The studies reviewed suggest that although models with more variables often demonstrate superior predictive performance, their quality and clinical relevance are more crucial than their quantity. Models like that used by Kim et al. [51], which included up to 86 variables, achieved an AUROC of 0.95, highlighting the importance of integrating diverse and dynamic data such as vital signs and medical device usage. Similarly, Šín et al. [60] achieved an exceptional AUROC of 0.99 by incorporating a comprehensive set of variables. Conversely, models with fewer variables, such as that used by Hyun et al. [55], reported lower performance (AUROC of 0.74). This illustrates the limitations of static or overly general variables, which fail to capture the dynamic nature of ICU patient conditions.

However, many variables can introduce challenges, including overfitting, increased computational complexity, and reduced interpretability, particularly in small datasets. Studies like that by Ladíos-Martin et al. [21], with 23 well-curated variables, demonstrated that selecting carefully relevant data can rival more complex models.

The study by Alderden et al. [58] demonstrated that including a larger set of variables did not significantly improve the predictive performance of models for HAPIs. Specifically, models developed using a parsimonious set of five easily accessible variables from EHRs performed almost as well as those using larger datasets, including variables from routine care and the Braden scale. This finding suggests that predictive accuracy can be achieved with a small, focused set of variables, reducing the complexity and burden of data collection. The study highlights the clinical feasibility of implementing streamlined models, which are more practical for real-time risk assessment in critical care settings.

Notwithstanding these achievements, obstacles persist. Most models were assessed retrospectively and lacked clinical validation, limiting their applicability. Furthermore, the heterogeneity of datasets and the absence of standardized performance metrics across studies render direct comparisons challenging. Future research should focus on the prospective validation of these models in multicenter studies, incorporating diverse patient populations and clinical settings. Furthermore, while AI models have demonstrated superior predictive power, their integration into clinical practice should consider the balance between advanced accuracy and the simplicity and accessibility of conventional tools like the Braden, Norton, CALCULATE, and EVAR-UCI scales, ensuring that predictive advancements translate into practical and meaningful improvements in patient care.

### 4.2. Key Variables in Prediction Models

AI-based prediction models for PIs demonstrate a significant advantage over traditional risk assessment tools by integrating variables that offer both breadth and specificity in predictive power. A review of the most significant variables used for model training in various studies highlights their pivotal role in enhancing model accuracy and applicability to ICU patient populations.

The most commonly identified variables are albumin, hemoglobin, BMI, glucose, and creatinine, which reflect the patient’s nutritional, inflammatory, and metabolic states. For instance, Alderden et al. [53] emphasized the importance of variables like surgical time, creatinine, and lactate, alongside physiological measures such as the Glasgow Coma Scale (GCS) and oxygenation levels (SpO_2_ < 90%), in predicting PI risk. These findings align with other studies, such as that by Cramer et al. [54], where mean arterial pressure, albumin, and ICU-specific factors like pressure-reduction device use were central to model performance.

Dynamic variables evolve throughout the patient’s ICU stay and further distinguish AI models from static tools. For example, Cho et al. [32] identified the importance of hemodynamic status, ventilator mode, and systolic blood pressure in improving predictive accuracy. By incorporating real-time updates to these variables, AI models can effectively capture temporal changes in patient health, such as sudden hemodynamic instability, that static tools like the Braden scale might overlook.

Nursing-related assessments also emerged as critical components of several models. The Braden scale and its subscales—such as mobility, friction/shear, and moisture—were frequently highlighted as key predictors. For example, Kaewprag et al. [33] and Vyas et al. [57] demonstrated how integrating traditional subscale scores into ML algorithms might enhance predictive outcomes. Similarly, Kim et al. [51] identified repositioning practices and Braden subscale scores like mobility, nutrition, and friction/shear as significant contributors to model accuracy. However, these variables are not consistently reported across studies, suggesting opportunities for the broader integration of care-related data.

Certain studies also highlighted unique predictors tailored to specific clinical contexts. For instance, Šín et al. [60] identified variables like glucose, albumin, and ICU length of stay as among the most critical in their high-performing random forest model (AUROC: 0.994). Similarly, Alderden et al. [58] identified the presence of skin tears, thin epidermis, and vasopressor infusion doses as highly predictive variables in their study, emphasizing the importance of nuanced skin assessments and the role of some medications.

Despite the success of these models, gaps remain in the consistent inclusion of variables directly related to prevention practices, such as frequency of repositioning, support surface use, and comprehensive skin care interventions. While some studies, like those by Kim et al. [51] and Alderden et al. [58], included data on repositioning, the limited application of this across models underscores a missed opportunity to enhance predictions with actionable nursing interventions. Moreover, although some studies were conducted in surgical ICUs [32,53,56,58], only a few explicitly incorporated perioperative variables into their predictive models. This omission is particularly significant given that surgery-related PIs can manifest within hours or up to three days postoperatively [62,63]. Therefore, the integration of perioperative risk factors, including surgery duration, type and duration of anesthesia, patient positioning, and intraoperative hemodynamic stability, could further improve the accuracy and clinical relevance of AI-based PI prediction in surgical ICU populations [62,63].

### 4.3. Clinical Implementation and Implications

The clinical implementation of these AI models has already shown promising results. Cho et al. [32] provided compelling evidence that integrating a Bayesian network model into clinical workflows resulted in a tenfold decrease in HAPIs and a 33% reduction in ICU length of stay. These findings highlight AI’s significant impact on patient outcomes and hospital resource management. Recent guidelines on PI prevention emphasize that clinical judgment is central to determining risk, yet clinicians face significant cognitive load with dozens of risk factors identified [6]. AI tools can reduce this burden by synthesizing complex data and providing actionable insights that align with evidence-based practices, enabling more consistent and informed decision-making.

Nevertheless, clinical adoption remains a challenge. As seen in the extraction data, the variability in model performance across different ICUs underscores the need for models to be tailored to the specific patient populations and care settings in which they are deployed. For example, Kim et al. [51] demonstrated the performance of deep learning-based models in multi-center settings, while studies like that by Alderden et al. [58] reported lower performance in single-center studies. This variation may be due to differences in patient demographics, treatment protocols, and data quality across institutions.

Moreover, while the review highlights successful implementations, such as the integration of AI tools into EHR systems, there is a lack of large-scale, multi-center studies that evaluate the long-term impacts of these models on clinical practice. Most studies report on the development and validation of models, but do not provide robust data on their practical utility.

While AI models show promise, their clinical adoption faces several challenges. Model performance variability across different ICUs highlights the need for tailored implementation strategies that account for specific patient populations, treatment protocols, and institutional workflows. Additionally, integrating AI models with EHRs and Clinical Decision Support Systems is crucial for practical application. User-friendly interfaces, automated alerts, and real-time risk assessments are essential to ensuring these tools are accessible and actionable for clinicians. However, the success of such integrations depends on clinicians’ trust in the system and their ability to interpret its outputs effectively [64].

Ethical considerations, such as bias in algorithms and accountability for AI-driven decisions, remain critical concerns. Hassan and El-Ashry [65] emphasized the importance of using diverse and representative training datasets to minimize biases and promote fairness. Additionally, maintaining the clinician’s role in decision-making is essential to preserve patient autonomy and trust. AI should serve as a tool to augment, not replace, clinical judgment.

Human factors, including user experience and workflow integration, also play a significant role in determining the success of AI adoption. Transparent models that provide clear explanations for their predictions are more likely to be trusted and used effectively by clinicians. One approach to addressing this challenge is the implementation of tools like feature importance analysis or visual interpretability methods, such as SHAP (SHapley Additive ExPlanations, version 0.47.1), as these tools enhance interpretability by showing how individual features influence predictions [51,59]. Ongoing education and collaboration among healthcare teams can further support the integration of AI tools into clinical practice. Furthermore, as suggested by Khosravi et al. [66], incorporating training programs and ongoing education for healthcare teams can strengthen interdisciplinary collaboration and enhance the usability of AI systems. Engaging clinicians in the model development process can also foster trust and ensure the tools meet the practical needs of clinical workflows.

Additionally, addressing systemic barriers, such as resource constraints and institutional readiness, is critical for successful AI integration. Evidence from previous AI applications in healthcare suggests that aligning organizational goals with AI capabilities is essential to maximize the impact of these technologies on patient outcomes and resource optimization [66].

Beyond organizational alignment, practical and economic barriers can significantly hinder the large-scale adoption of AI tools in clinical settings. These include the need for compatible infrastructure, integration with existing electronic health record systems, and secure mechanisms for data storage and transfer. Furthermore, healthcare institutions must invest in workforce training and ongoing technical support to ensure the proper use and oversight of AI tools. These requirements can be especially challenging in resource-limited environments [67].

Economic constraints remain a critical issue. The development, validation, implementation, and maintenance of AI systems involve substantial financial investment, including software licensing, hardware acquisition, and system upgrades. Without dedicated funding and coordinated efforts among health systems, policymakers, and industry stakeholders, these barriers may delay or even prevent the equitable implementation of AI across different care settings [67,68].

### 4.4. Strengths and Limitations

A significant strength of this review is its comprehensive scope, encompassing a wide range of predictive models and study designs. This breadth allows for a thorough understanding of the landscape of PI prediction in the ICU setting using AI. However, several limitations should be acknowledged. Firstly, the heterogeneity among the included studies, in terms of patient populations, ICU settings, methods, and outcome measures, complicates direct comparisons and the synthesis of results. Secondly, many studies had small sample sizes or were conducted in single-center settings, potentially limiting the generalizability of their findings. Lastly, the rapid advancement in ML technologies may mean that some older studies do not reflect the current state-of-the-art in predictive modeling.

### 4.5. Future Research Directions

This review highlights several key areas for future research, as follows:External Validation and Multicenter Studies—Prospective, multicenter trials across diverse ICU settings are essential to ensure predictive models’ generalizability and real-world applicability. Diagnostic randomized controlled trials are critical for evaluating their true clinical effectiveness beyond retrospective metrics [69,70];Incorporation of Underutilized Variables—Future models should include repositioning frequency, support surface use, perioperative variables, skin care interventions, and staff workload metrics, alongside physiological and laboratory data, to improve specificity and relevance;Standardization of Metrics—Uniform reporting and performance metrics are needed to enable cross-study comparisons, enhance evidence synthesis, and improve model refinement;Real-Time Integration—Dynamic models with continuous monitoring, such as time-series analysis, should be integrated into EHRs for seamless, proactive care;Addressing Ethical and Human Factors—Models must be interpretable and user-friendly to foster trust and adoption among clinicians. Ethical concerns, such as algorithmic bias and transparency, must also be addressed to ensure equitable implementation.

## 5. Conclusions

This scoping review underscores the potential of predictive models to improve PI prevention in ICU patients. While traditional risk assessment tools remain valuable, ML models offer promising advancements. Nevertheless, further research is necessary to validate these models and ensure their effective integration into clinical practice. By leveraging predictive models and integrating them into clinical decision support systems, ICU teams can proactively identify high-risk patients and implement timely interventions. These algorithms can analyze patient data, such as vital signs, lab results, and risk factors, providing, in real-time, actionable alerts or recommendations to healthcare professionals. This creates opportunities for targeted preventive measures, optimized resource allocation, improved patient outcomes, and reduced healthcare costs by minimizing the incidence and severity of pressure injuries. However, practical challenges remain, including robust and interoperable data systems, the risk of alert fatigue among clinicians, proper staff training, and the need to validate and adapt predictive models to diverse clinical settings. Addressing these challenges is essential to fully realizing the potential of predictive models in routine ICU practice.

## Figures and Tables

**Figure 1 nursrep-15-00126-f001:**
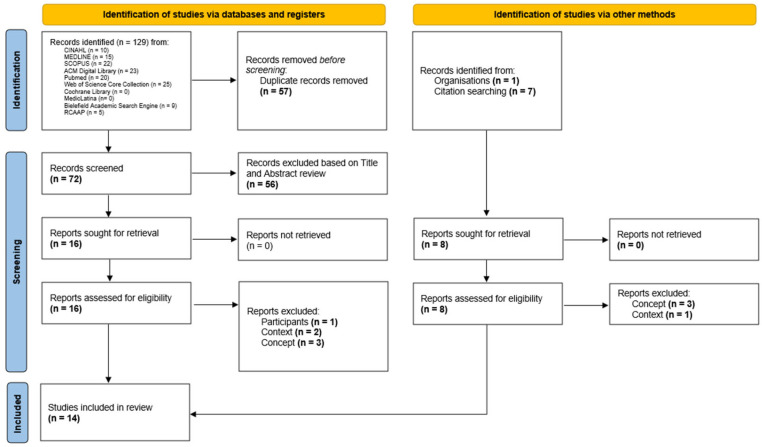
Flow diagram of the paper identification, selection, eligibility, and inclusion process [48].

**Table 1 nursrep-15-00126-t001:** Pressure injury prediction models by study and best-performing model.

Author	Prediction Models	Best Performing Model
Cho et al. 2013 [32]	Bayesian Networks	Bayesian Networks
Kaewprag et al. 2015 [52]	Logistic Regression; Support Vector Machine; Decision Tree; Random Forest; k-nearest neighbors; Naïve Bayes	Logistic Regression
Kaewprag et al. 2017 [33]	Bayesian Networks	Bayesian Networks
Alderden et al. 2018 [53]	Random Forest	Random Forest
Cramer et al. 2019 [54]	Logistic Regression; Elastic Net; Support Vector Machine; Random Forest; GBM; Neural Networks	Logistic Regression
Hyun et al. 2019 [55]	Logistic Regression	Logistic Regression
Choi et al. 2020 [56]	Naïve Bayes	Naïves Bayes
Ladíos-Martin et al. 2020 [21]	Logistic Regression; Bayes Point Machine; Averaged Perception; Boosted Decision Tree; Boosted Decision Forest; Decision Jungle; Locally Deep Support Vector Machine; Neural Networks; Support Vector Machine	Logistic Regression
Vyas et al. 2020 [57]	XGBoost	XGBoost
Alderden et al. 2021 [58]	Neural Networks; Random Forest; GBM; AdaBoost; Logistic Regression	GBM
Alderden et al. 2022 [59]	K-nearest neighbors; Logistic Regression; Multi-layer Perceptron; Naïve Bayes; Random Forest; Support Vector Machine	Ensemble SuperLearner
Šín et al. 2022 [60]	K-nearest neighbors; Logistic Regression; Multi-layer Perceptron; Naïve Bayes; Random Forest; Support Vector Machine	Random Forest
Ho et al. 2024 [61]	AdaBoost; Decision Tree; Logistic Regression; K-nearest neighbors; Multi-layer Perceptron; Random Forest; Support Vector Machine; GBM; MedaBoost	MedaBoost
Kim et al. 2024 [51]	RNN; GRU; LSTM; Logistic Regression; Decision Tree; Random Forest; XGBoost; GRU-D; GRU-D++	GRU-D++

Abbreviations: GBM—Gradient Boosting Machine; GRU—Gated Recurrent Unit; GRU-D++—Gated Recurrent Unit with a decay; LSTM—long short-term memory; MedaBoost—Medical Expert Disagreement Adaptative Boosting; RNN—recurrent neural network; XGBoost—eXtreme Gradient Boosting Machine.

**Table 2 nursrep-15-00126-t002:** Reported performance metrics for the best-performing prediction model in each study.

Author	Model	ACC	AUROC	SEN	SPE	PPV	NPV
Cho et al. 2013 [32]	Bayesian Networks	-	0.85	0.82	0.76	0.36	0.96
Kaewprag et al. 2015 [52]	Logistic Regression	-	0.83	0.16	0.99	0.56	0.93
Kaewprag et al. 2017 [33]	Bayesian Networks	-	0.83	0.46	0.91	0.29	0.95
Alderden et al. 2018 [53]	Random Forest	-	0.79	-	-	-	-
Cramer et al. 2019 [54]	Logistic Regression	-	-	0.71	-	0.09	-
Hyun et al. 2019 [55]	Logistic Regression	0.92	0.74	0.65	0.69	0.21	0.96
Choi et al. 2020 [56]	Naïves Bayes	-	0.82	0.6	0.89	0.23	0.98
		-	0.68	0.85	0.76	0.37	0.97
Ladíos-Martin et al. 2020 [21]	Logistic Regression	0.87	0.88	0.75	0.88	0.22	0.99
Vyas et al. 2020 [57]	XGBoost	0.95	-	0.84	0.97	0.87	0.97
Alderden et al. 2021 [58]	GBM	-	0.82	-	-	-	-
Alderden et al. 2022 [59]	Ensemble SuperLearner	-	0.81	-	-	-	-
Šín et al. 2022 [60]	Random Forest	0.96	0.99	0.92	-	0.95	-
Ho et al. 2024 [61]	MedaBoost	-	0.9	-	-	-	-
Kim et al. 2024 [51]	GRU-D++	-	0.95	-	-	-	-

Note: “-“ means not reported data. Abbreviations: ACC–accuracy; AUROC—area under the receiver operating characteristic curve; GBM—Gradient Boosting Machine; GRU-D++—Gated Recurrent Unit with a decay; MedaBoost—Medical Expert Disagreement Adaptative Boosting; NPV—Negative Predictive Value; PPV—Positive Predictive Value; SEN—sensitivity; SPE—specificity; XGBoost—eXtreme Gradient Boosting Machine.

## Data Availability

Not applicable.

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
