# Peer review of "Pressure Injury Prediction in Intensive Care Units Using Artificial Intelligence: A Scoping Review"

_nursrep, 2025, doi:10.3390/nursrep15040126_

Round 1
Reviewer 1 Report
Comments and Suggestions for Authors
Dear Authors
Thank you for your efforts. I believe that your research will contribute to the field and literature. I wish you ease in your process.
In order for this study to be more understandable for readers, artificial intelligence models can be given in more detail in the findings section. They have given them as main categories but have not explained how they do this. For example, is the data entered manually? Or is it automatically pulled when the patient's protocol or name is written? In addition, is there a system that visually scans the wound in wound care? Or is the stage entered manually?
Also, there were not many data on surgical pressure injuries in this study due to limitations? Various measurement tools such as Norton, Waterlow, Braden were mentioned, but it is not recommended to use separate scales in the literature for surgical pressure injuries. Some scales developed for surgical pressure injuries; there is a scale developed by 3S, Munro, for example. I think surgical pressure injuries are also important because most patients in intensive care can stay in intensive care after invasive or surgical interventions. If there is no data on this in the studies they examine, I suggest that this should be added and given as a suggestion for future studies.
Author Response
Dear Reviewer,
Thank you for your time and thoughtful evaluation of our manuscript. Please see the attachment for our detailed, point-by-point response to your comments, prepared according to the suggested template.
We greatly appreciate your valuable feedback, which has helped us to improve the quality and clarity of our work.
Kind regards,
José Alves (On behalf of all authors)

Reviewer 2 Report
Comments and Suggestions for Authors
Dear author, I thought this was a very fluent article while reading it. I think it will contribute to the field.
-“The development of 36 a PI can” s suffix is missing (Line 37)
-Commas in numerical values should be replaced with (,) dots (.) both in the text and in the table.
Author Response

(The authors gave the same response as above.)

Reviewer 3 Report
Comments and Suggestions for Authors
The manuscript addresses a highly relevant and innovative topic and the initiative to explore the application of AI models.
However, to further improve its quality, the authors should consider clarifying several additional points:
- The potential influence of clinician experience and inter-observer variability in the use of traditional scales.
- Practical and economic barriers to implementing AI tools in different clinical settings.
Addressing these issues will strengthen the clinical applicability and overall clarity of the manuscript.
Author Response

(The authors gave the same response as above.)

Reviewer 4 Report
Comments and Suggestions for Authors
This study written for the prediction of pressure injury in intensive care units using artificial intelligence looks pretty well-equipped but would look much better with a few minor tweaks.
- Please expand the conceptual framework in the introduction.
- Include a strong paragraph supporting your purpose statement.
- Include your research questions clearly in the study
Author Response

(The authors gave the same response as above.)
